# Opportunistic Survey Analyses Reveal a Recent Decline of Skate (Rajiformes) Biomass in Falkland Islands Waters

Andreas Winter * and Alexander Arkhipkin

Falkland Islands Fisheries Department, Bypass Road, Stanley FIQQ 1ZZ, Falkland Islands
* Correspondence: awinter@naturalresources.gov.fk

**Abstract:** Data were combined from surveys in 2013, 2018, 2019 and 2021 to examine biomass changes of the skate (Rajiformes) stock in waters around the Falkland Islands. The 2013 and 2019 surveys were research surveys for skate abundance and used 90 mm trawl mesh to capture species and size compositions. The 2018 survey was a mesh size trial, and the 2021 survey was a commercial exploration; both used 400 mm trawl mesh to target skates. All four surveys were compared for biomass by deselecting the proportions of skates per size interval caught in 90 mm mesh that would not have been caught in 400 mm mesh, calculated with the SELECT method. Estimated commercial-size skate biomass decreased for most species individually and approximately 61% overall. Estimated commercial-size skate biomass also decreased by as much south of 51° latitude, the area that was closed to skate target fishing since 1996, as north of 51° latitude, which has been maintained open to skate target fishing. The outcome is correlated with continuing skate bycatch in finfish trawls.

**Keywords:** biomass estimation; mesh size; Falkland Islands; skates; species assemblage





## 1. Introduction

The Falklands Islands fishing zone comprises a high biomass and diversity of skate species (Rajiformes) [1–4], and has provided one of the globally few targeted commercial fisheries for skates [5,6]. Targeted skate catches have ranged as high as 3500–4400 tonnes per year, with 10–12 vessels participating. As a conservation measure, targeted fishing on skates in the Falkland Islands was closed south of 51° S in 1996 [1]. Directed fishing for skates continued to be open north of 51° S, but has had little operation in the past few years (i.e., 2017–2022; [7]), especially following a regulatory implementation of 400 mm minimum trawl mesh to reduce bycatch of undersized individuals and non-target species starting in 2021 [8]. Given the low input, a commercial catch/effort-based stock assessment of skates has not been a priority to perform since 2018 [9].

Catch/effort-based stock assessment has also been constrained by the absence of species identification in the fishery. The Falkland Islands skate assemblage includes at least 15 species, but commercial vessels are only required to report a single aggregated code for skates [2,5], presenting a risk that some individual species could decline undetected within the multispecies assemblage [10]. The risk is exacerbated for species with older ages of maturation [2]. Furthermore, skate bycatches are taken regularly in finfish trawl fisheries, but these bycatches cannot be reliably indexed to skate abundance given the shifting bases of the main targets of finfish trawls [7].

Monitoring of skate stocks has therefore been available primarily through periodic trawl surveys covering known skate habitat areas. Recent indications of renewed interest by skate licence quota holders [8] have prioritized estimating skate biomass and population trends from these surveys. However, the surveys had different tasks rather than being focused on biomass estimation. In this paper we describe approaches that were implemented to standardize data between surveys, based on the principle of differential catch size selectivity between trawl meshes [11,12]. Biomass changes from standardized data are

compared with effort trends over time in the commercial fisheries. Additional to changes over time, biomass trends are compared between south of 51° S in the Falkland Islands fishing zone (where targeted skate fishing has been closed since 1996) and north of 51° S in the Falkland Islands fishing zone (where targeted skate fishing has remained open).

The analysis of these surveys provides an updated status of the Falkland Islands multispecies skate assemblage. Information from this study is critical to evaluate the continuing sustainability of skates in a commercial fishery, and draw conclusions on the utility of a skate target exclusion zone.

## 2. Materials and Methods

Four surveys are included in this analysis (Table 1); conducted in 2013, 2018, 2019, and 2021 during October/November, the months of highest skate abundance around the Falkland Islands [3]. The 2013 and 2021 surveys trawled both north and south; the 2018 and 2019 surveys north only. The 2013 and 2019 surveys were Falkland Islands Fisheries Department (FIFD) research cruises and used standard bottom trawl with 90 mm mesh in the codend for its high capture retention. The 2018 survey was a trawl mesh trial comparing 110, 300, and 400 mm meshes in the codend. The 2021 survey was an industry-sponsored exploratory trip to examine the current cost-effectiveness of skate fishing, and used regulatory 400 mm mesh in the codend. In each survey, skates were identified to species, weighed, and measured across disk width to the lowest cm.

**Table 1.** Summary of skate trawl surveys, corresponding to Figure 1.

| Year | Mesh (mm) | Trawls (N) | | Reference |
|------|-----------|------------|-------|-----------|
| | | **North** | **South** | |
| 2013 | 90 | 30 | 32 | Pompert et al., 2014 [13] |
| 2018 | 400 | 6 | 0 | Arkhipkin et al., 2018 [14] |
| 2019 | 90 | 56 | 0 | Goyot et al., 2020 [15] |
| 2021 | 400 | 7 | 12 | Parkyn et al., 2022 [16] |

To establish an equal baseline for comparing skate biomasses between surveys, in 2018 only the 400 mm mesh catches were analysed. In 2013 and 2019, 90 mm mesh catches were thinned by deselecting smaller skates in proportion to their likelihood of not being caught in 400 mm mesh. These proportions of likelihood were calculated using the SELECT (Share Each Length class's Catch Total) method for unpaired trawl data [11,12], which fits a logistic model to the numbers per size interval retained in the larger of two meshes being compared, vs. the numbers per size interval of both meshes. For example, if the SELECT logistic model predicted that 30% of skates of a given disk-width occurred in the 400 mm mesh, then 70% of skates of that disk width were deselected from 90 mm mesh trawls. Other than trawl meshes, all four surveys used the same bottom trawl gear configuration of ground rope and tickler chain.

Skate density per species per trawl was calculated as the catch weight (or deselected 400 mm—equivalent catch weight) divided by swept area (trawl width × trawl distance), where trawl distance is the straight-line trajectory from the start position to end position recorded for each trawl. Skate density was then multiplied by swept area and extrapolated to estimated total biomass in the north and south "Skate Boxes" (Figure 1). The North Skate Box comprises 36,515 km$^2$ of the northern slope that has been the primary area for commercial skate fishing [15]. The South Skate Box comprises 22,559 km$^2$ that is representative skate habitat across the southern slope and south-eastern outer shelf [4,13]. Both Skate Boxes were partitioned into 5 × 5 km$^2$ grids, and extrapolation calculated using inverse distance weighting, a comparatively simple procedure suitable for small data sets without defined spatial structure [17]. The southern slope and south-eastern outer shelf zones of the South Skate Box were considered a habitat continuum.

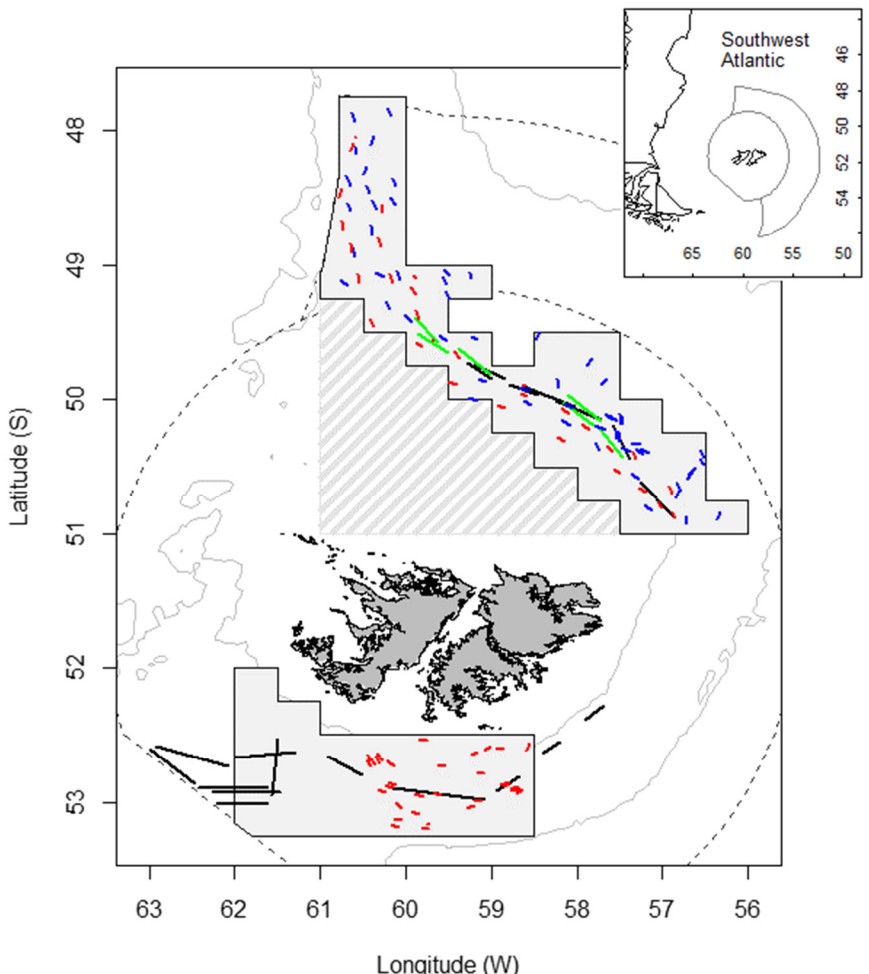

**Figure 1.** Distribution of survey trawl tracks in 2013 (red), 2018 (green—400 mm mesh only), 2019 (blue), and 2021 (black); corresponding to Table 1. Grey shaded areas: North and South Skate Boxes. Grey striated area: shelf skate spawning area. Boundary lines: Falkland Islands Conservation Zones.

Variability of biomass estimates in each year, per species, in either North or South Skate Box, was calculated by randomly re-sampling and re-combining the different data components used to derive biomass estimates: (1) randomly re-sampling from the normal distribution the proportion that skate represented in the total catch of each trawl (as opposed to other bycatch), (2) randomly re-sampling with replacement those skates that would make up that proportion per trawl (and if 90 mm mesh; the outcome that any skate would be deselected from equivalence to 400 mm mesh by double bootstrapping [12]), and (3) randomly re-assigning the trawl catch location between start point and endpoint of each track, for inverse distance weighting. Randomizations were iterated 10,000× in either Skate Box in each year. Changes in biomass from 2013 to 2021 were modelled in the north with generalized additive models (GAM) re-computed from each randomized iteration, using years as the predictor variable, and in the south (comprising only two years of data) were compared with box plots. Statistical significances of changes in biomass from 2013 to 2021 were calculated as the proportions of the 10,000 randomizations that produced a lower biomass GAM prediction (north) or iteration (south) in 2021 than in 2013. Statistical significances of difference between north and south in biomass changes from 2013 to 2021 were calculated as the proportions of the 10,000 randomizations for which the north change was lower than the south change. Statistical analyses were carried out in R version 4.0.5 [18]; GAMs were calculated in R package mgcv [19].

Changes in biomass were also compared to the trends in bottom-trawl effort over time covering the areas of the North and South Skate Boxes, and of the major spawning area for

shelf skates north of the Falkland Islands (Figure 1; [3]). Bottom-trawl effort was defined as the number of vessel-days per year under finfish licence or skate licence from mandatory daily catch reports to the FIFD, and examined as a proxy for the total potential fishing impact on the skate stock. Finfish licence was included because skate catches in Falkland Islands waters are largely taken as bycatch [7,9], as in fishing regions elsewhere [20–22]. Daily catch reports include entries for discard as well as retained catch, but discard records are of questionable reliability when not backed by observer coverage, which is relatively low in these fisheries. Bottom-trawl effort trends across years were fit to a LOESS function (span = 1, degree = 2).

## 3. Results

Ten skate species were identified in the four surveys and compared (Table 2). Additional skate species *Bathyraja magellanica* (Philippi, 1902), *Dipturus trachydermus* (Krefft & Stehmann, 1975), and *Psammobatis* spp. were recorded in small quantities (<2.3% of total skate catches) in the 2013 and 2019 surveys, but not the 2018 or 2021 surveys, and therefore not analysed as deselection equivalences from 90 mm to 400 mm mesh could not be computed. For all other species the SELECT method found statistically significant size distribution differences between 90 mm and 400 mm mesh, although with large variation in the precision of logistic models (Figure 2). Total biomass deselection equivalences ranged from 0.6% (*Dipturus lamillai* Concha, Caira, Ebert & Pompert, 2019, 2013—south) to 83.2% (*Bathyraja albomaculata* (Norman, 1937), 2019—north).

**Table 2.** Summary of skate species (with FI species codes) captured in the surveys, percentages of biomass change from 2013 to 2021, statistical significances of change within north and south Skate Boxes as well as of the difference between north and south. NS: not significant; $p > 0.05$.

| | Species | Biomass change 2013 to 2021 | | | | Change Diff. between N and S |
|---|---|---|---|---|---|---|
| | | N | | S | | |
| | | % | Sig. | % | Sig. | Sig. |
| RAL | *Bathyraja albomaculata* | −58.8 | NS | −60.3 | $p < 0.050$ | NS |
| RBR | *Bathyraja brachyurops* | −48.3 | $p < 0.050$ | −98.6 | $p < 0.001$ | $p < 0.001$ |
| RBZ | *Bathyraja cousseauae* | −47.7 | NS | −65.4 | NS | NS |
| RDA | *Dipturus argentinensis* | +82.9 | NS | +Inf | $p < 0.001$ | $p < 0.001$ |
| RDO | *Amblyraja doellojuradoi* | −92.2 | $p < 0.010$ | −65.0 | $p < 0.050$ | NS |
| RFL | *Dipturus lamillai* | −39.4 | $p < 0.050$ | −10.9 | NS | NS |
| RGR | *Bathyraja griseocauda* | −65.0 | $p < 0.050$ | −1.1 | NS | NS |
| RMC | *Bathyraja macloviana* | −80.2 | $p < 0.005$ | −97.2 | $p < 0.001$ | $p < 0.050$ |
| RMU | *Bathyraja multispinis* | −24.3 | NS | −22.2 | NS | NS |
| RSC | *Bathyraja scaphiops* | −86.6 | $p < 0.001$ | −79.4 | $p < 0.001$ | NS |
| All | | −55.7 | $p < 0.001$ | −69.1 | $p < 0.005$ | NS |

Estimated commercial-size biomasses of most skate species, and the aggregates of all skate species, decreased significantly from 2013 to 2021, both north and south (Table 2, Figures 3 and 4). One species (*Dipturus argentinensis* Diaz de Astarloa, Mabragaña, Hanner & Figueroa, 2008) was not recorded in the south in the 2013 survey, but recorded in the south in the 2021 survey, representing the only case of a statistically significant increase over time. While *D. argentinensis* was consistently one of the least abundant skates in all surveys (Figures 3 and 4), increase over time also occurred in the north surveys, over the latter part of the time series (Table 2, Figure 3), and appears substantiated from commercial fisheries observer data in the south, which showed a significantly increasing LOESS trend of *D. argentinensis* kg per sampled station between 2010 and 2021.

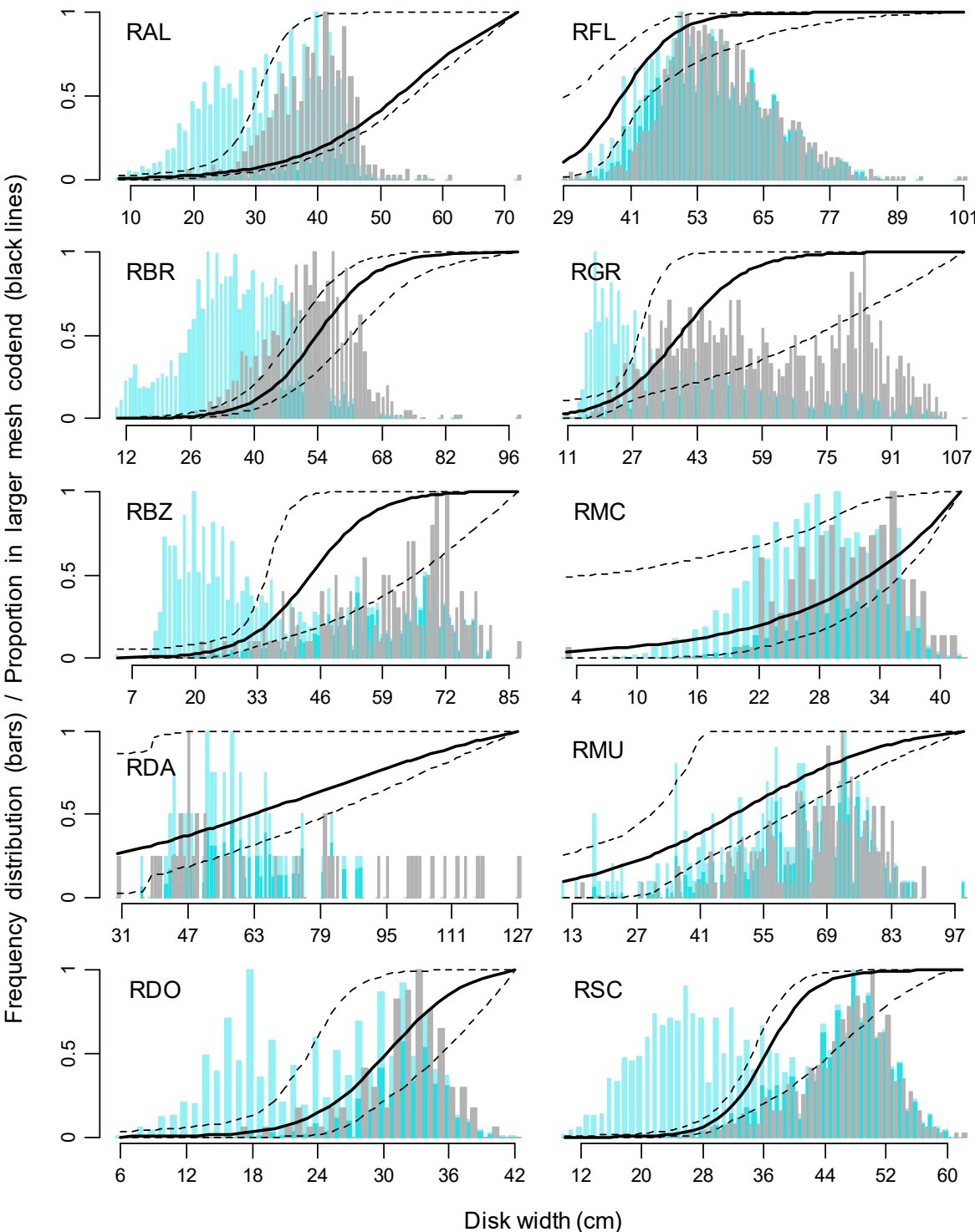

**Figure 2.** Combined disk width frequency distributions in 90 mm mesh (blue) and 400 mm mesh (grey) per skate species. Lighter sections of the blue bars: fractions per disk width interval deselected to equalize 90 mm mesh catches to 400 mm mesh catches. Black lines: SELECT logistic models of proportions retained in the larger mesh ± 95% confidence intervals. Species codes correspond to Table 2.

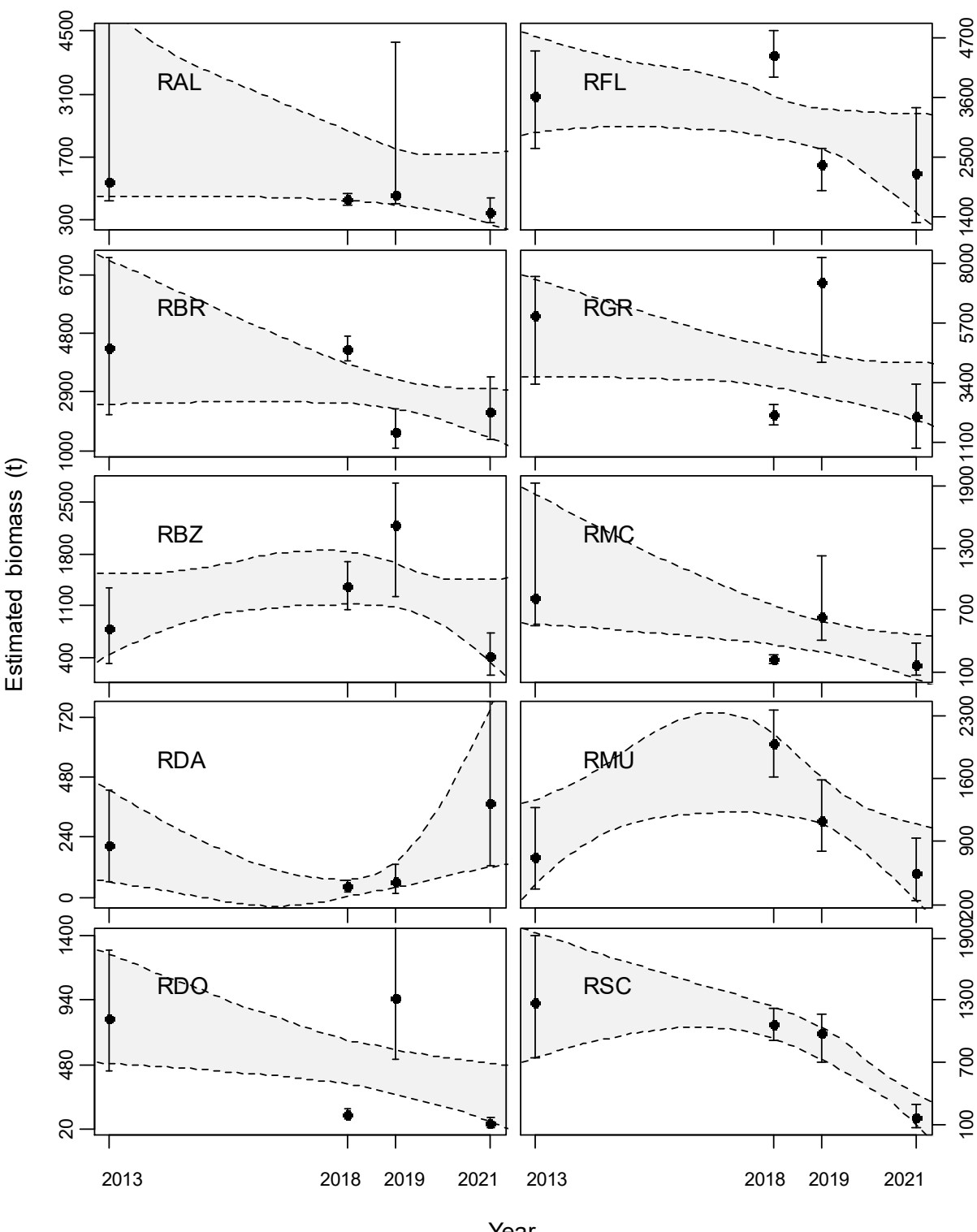

**Figure 3.** Biomass estimates of skate species in the North Skate box, from the four surveys in 2013 to 2021. Black symbols per year: point estimate ± 95% confidence intervals. Broken lines and grey under-shading: 95% confidence interval of the inter-annual GAM from randomized resampling. Falkland Islands species codes correspond to Table 2.

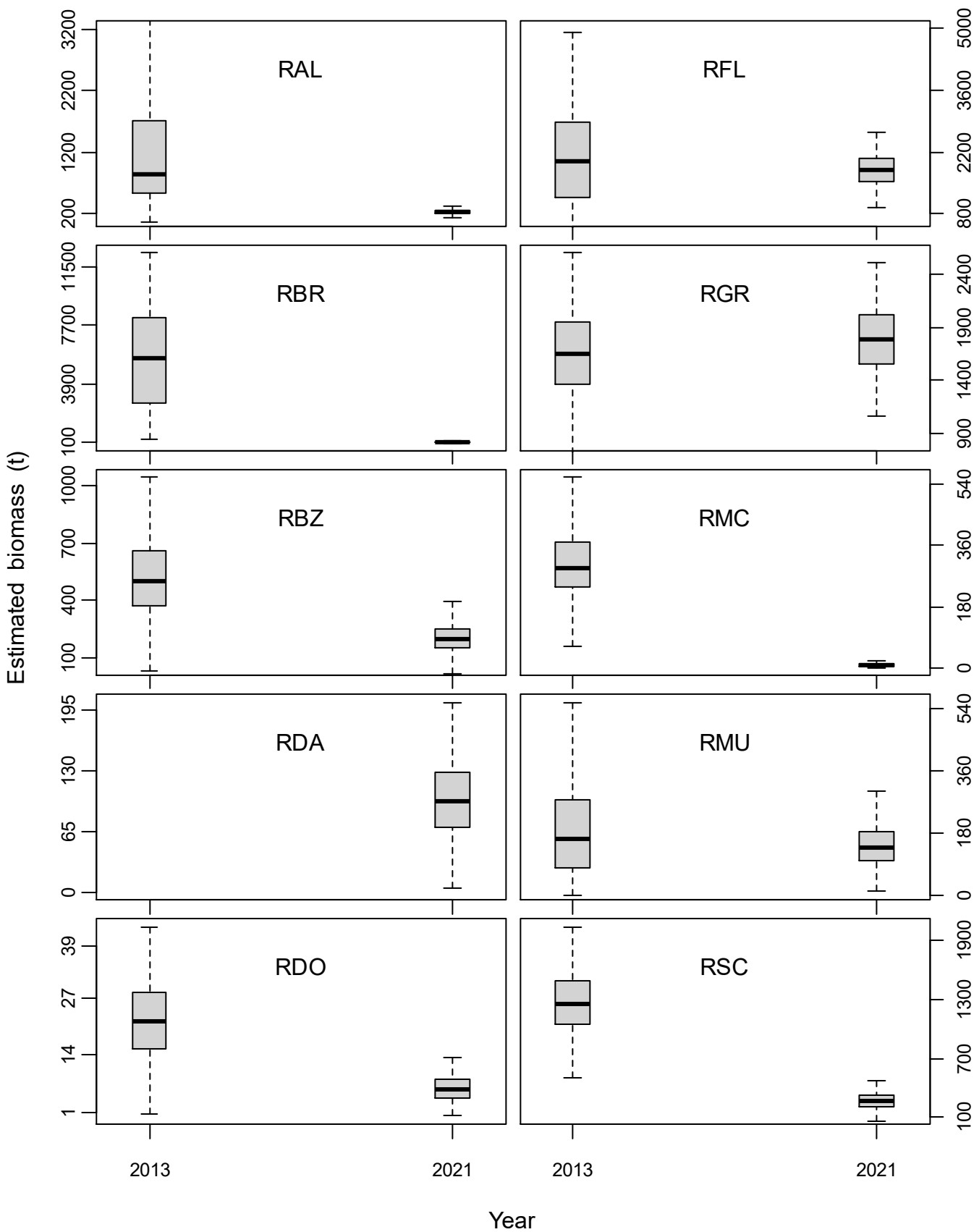

**Figure 4.** Biomass estimates of skate species in the South Skate box, from the two surveys in 2013 and 2021. Box plots: first, second, and third quartiles, whiskers to approximate 95% confidence intervals from randomized resampling in either year. Falkland Islands species codes correspond to Table 2.

Among the nine skate species other than *D. argentinensis*, five species had a higher proportional biomass decrease, from 2013 to 2021, in the north than south, and four species had a higher proportional biomass decrease in the south than north. However, the north–south difference was statistically significant ($p < 0.05$) for only two species, *Bathyraja brachyurops* (Fowler, 1910) and *Bathyraja macloviana* (Norman, 1937), both of which had more decrease in the south than in the north. The north–south difference in decrease was not statistically significant for the aggregate biomass among all skate species (Table 2). The aggregate north and south decrease for all species was 61% (95% confidence interval: 45% to 70%).

Bottom-trawl effort followed parallel time series trends in the North and South Skate Boxes, as well as the shelf spawning area: increasing sharply from 2008 to 2010–2012, then decreasing from about 2016 (Figure 5). Finfish licences have had seasonal restrictions, but no explicit area exclusion like skate licences from south of 51° latitude after 1995. The increase of finfish licence effort in the south in 2016 and 2017 (Figure 5) was driven by a brief phase of commercial interest in grenadier (*Macrourus* spp.).

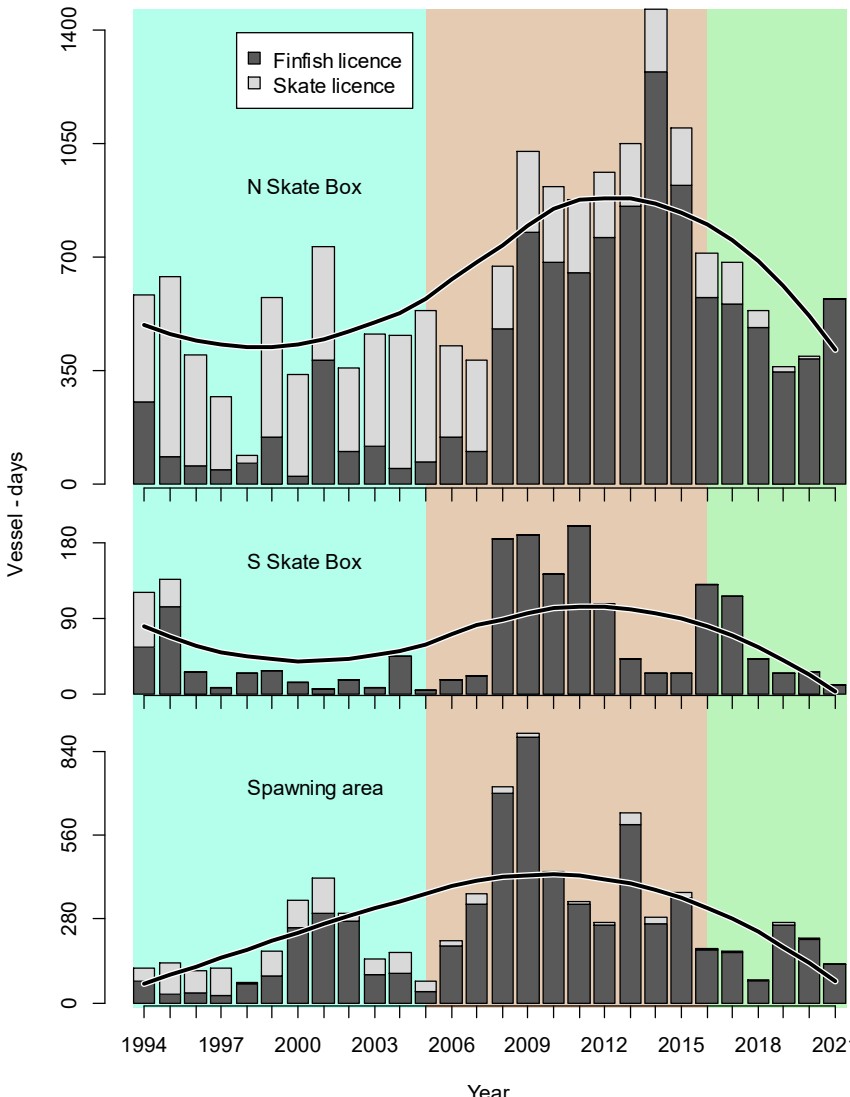

**Figure 5.** Fishing vessel-days per year reported in the area of the North Skate Box, South Skate Box, and shelf skate spawning area, under finfish licence and skate licences. Blue, brown, and green shaded areas: periods when the finfish assemblage and finfish catches were dominated by southern blue whiting, rock cod, and common hake, respectively. Black lines: LOESS smooth of total fishing vessel-days.

## 4. Discussion

Skate catches in the four surveys included two species listed as near-threatened—*B. brachyurops* [23] and *B. macloviana* [24], one species listed as vulnerable—*B. albomaculata* [25], and one species listed as endangered—*Bathyraja griseocauda* (Norman, 1937) [26]. *B. brachyurops*, *B. albomaculata*, and *B. griseocauda* are three of the four large species that have dominated Falkland Islands skate fishing [2]. The fourth large species, *D. lamillai*, was recently described as a new species [27] resident around the Falkland Islands and listed as least concern [28], which has separated this new species from the endangered status of its previous conspecific *Dipturus chilensis* (Guichenot, 1848) [29]. All of these species showed significant commercial-size biomass decreases in one or both of the North and South Skate Boxes between 2013 and 2021, presenting a contrast to [4], which found that abundance trends of major skate species did not decrease from 1994 to 2013.

The pattern may reflect broader changes in the Falkland Islands trawl fisheries. Until 2005, finfish catches were dominated by the pelagic planktivore southern blue whiting *Micromesistius australis* (Norman, 1937) [30]. However, from 2006 onwards, southern blue whiting was rapidly overtaken by the near-bottom browser rock cod *Patagonotothen ramsayi* (Regan, 1913) [31]. Rock cod inhabits shelf and shelf-edge areas, thus overlapping with a majority of skates, whereas southern blue whiting is mainly an upper continental slope species [32–35]. The increased bottom-trawl fishery target of rock cod would therefore take more skate bycatch when the vessels started fishing the same shelf areas within a few years. Large skates in Falkland Islands waters attain maturity at a minimum of 8–10 years [2,36,37], suggesting that the change impact on skate abundance from a southern blue whiting dominated finfish assemblage to a rock cod dominated finfish assemblage was lagged by about one generation.

Quantitative results of this analysis should be considered approximate given the differences among surveys. For example, the 90 mm surveys in 2013 and 2019 towed much shorter tracks than the 400 mm surveys in 2018 and 2021 (Figure 1), to reduce saturation of the net. Trawl durations are known to have a potential influence on catch, with shorter trawls often producing relatively higher CPUE [38–40], although an interactive effect of trawl duration with mesh size has not been explicitly studied. Conversely, the longer tracks in 2018 and 2021 compensated for their lower numbers of trawls, bringing the total survey distances covered much closer to parity with 2013 and 2019. The multiple factors included in randomizations served to standardize quantitative results between surveys in this study. Some part of inter-annual change in abundance might also be caused by distributional shifts, as skates have seasonal migratory behaviour [2,3], and underscores the importance of continuing to monitor skate bycatch in all fisheries.

Despite limitations, opportunistic uses of varying surveys are recognized as an asset to fisheries assessment when alternative data sources are lacking [41,42]. In Falkland Islands waters, the combination of four surveys over a span of nine years (2013–2021) demonstrated changes of the skate population with an estimated commercial-size biomass reduction on the order of 61% overall, and significant reductions in eight of ten species individually despite high within-year uncertainty among several of the surveys. Notably, skate abundances overall did not decrease less in the south part of the fishing zone, which has been closed to commercial skate fishing since 1996, than in the north part of the Falkland Islands fishing zone, which has remained open, indicative that the target skate fishing exclusion was ineffective as long as other trawl fisheries continued to be allocated over the same area at a high rate. However, trends may foreseeably improve for skates. Since 2016, dominance of the Falkland Islands finfish assemblage has changed again, from rock cod to the nektonic predator common hake *Merluccius hubbsi* Marini, 1933 [7]. Common hake is fished mainly in the northern and western parts of Falkland Island waters [43], resulting in decreased trawl effort in the skate habitat areas (Figure 5). With precautionary management such as area restrictions on bottom trawling, a slow recovery of the skate stocks may be anticipated.

**Author Contributions:** A.W.: conceptualization, methodology, analysis, writing—original draft preparation, writing—review and editing; A.A.: conceptualization, investigation, writing—review and editing. All authors have read and agreed to the published version of the manuscript.

**Funding:** This research received no external funding.

**Institutional Review Board Statement:** Ethical review and approval were waived for this study, due to survey trawls being conducted with commercial trawl gear under Falkland Islands Government experimental licence. Fish arrived on-board dead, and were used for product following study data measurements.

**Data Availability Statement:** Data used in this study are available on reasonable request from the Falkland Islands Fisheries Department.

**Acknowledgments:** We are grateful to the officers, crew, and operational managers of the fishing vessels *Castelo*, *Petrel*, and *Monteferro*, for their collaboration in carrying out these surveys. We also thank the Fisheries Department scientific staff and observers who participated in survey sampling, and data collection and preparation. The article was greatly improved through constructive comments by three anonymous reviewers.

**Conflicts of Interest:** The authors declare no conflict of interest.

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
