# Peer review of "Opportunistic Survey Analyses Reveal a Recent Decline of Skate (Rajiformes) Biomass in Falkland Islands Waters"

_fishes, doi:10.3390/fishes8010024_

Round 1
Reviewer 1 Report
In this manuscript the authors aimed to demonstrated biomass trends over time for the skate assemblage in the Falkland Islands. They used surveys over the 2013 - 2021 period, standardizing the surveys to compensate for the differences in mesh sizes used. Outcomes describe a downward trend for most of the species. In this context, the authors have tried to make use of the best available data to explore and evaluate the status of these (data-limited) species. Such analysis is highly needed to ensure appropriate management action can be taken to ensure sustainably exploitation of these stocks. That being said I do have some issues with the analysis performed:
1) I could not fully understand the methods applied. I think the manuscript would benefit from a step-by-step description. Some methods have been aggregated in a single sentence, which make it highly unclear. I would suggest revisiting the methods section or add supplementary section.
2) Results section is very brief. I think more explanation is required (see comments in the attached pdf). Please have a clear look at table 2 in the results. I would suggest changing decrease in biomass to a change biomass. As such a decrease could be noted with a minus. This prevents misinterpretation of the -82%.
3) My main concern are the analysis and conclusions for the Southern skate box. The authors compare 2 points in time. There is no reference on where the population is in between 2013 and 2021. As the surveys are not specifically designed to catch skates (if understood correctly) something else could have affected catchability in a specific year. This is also seen in the North Skate box, where annual fluctuations are clearly observed for some species, i.e. there is no clear trend (e.g. RGR). How certain are the authors the populations are actually declining compared to whether there are other effects influencing skate catches in the survey? It’s not because you don't catch them, they are not there. Would a comparison between a commercial CPUE plot and the CPUE in the survey be useful to see if there is a similar trend over time? Or add information from observers as is mentioned but not shown for RDA in line 140.
Overall, I think the paper is well written and requires minor changes.
Reviewer 2 Report
Please add authority for each species the first time you mention them.
Reviewer 3 Report
Page 2, line(s) 49: Please synthesize all relevant background statements and write a paragraph detailing the study's importance and critical purpose.
Page 2, line(s) 66: Please specify the full name of SELECT.
Page 2, line(s) 66-72: The principle of the SELECT method should be explained in the introduction.
Page 2, line(s) 73-77: The trawling process may cross more than two grids. How do you define the geographic information of the samples?
Page 2, line(s) 77-81: The study area contains different habitats. Any statistical evidence supporting the feasibility of the error produced by inverse distance weighting?
Page 3, Figure 1: The information in the picture is too complicated. Please clear the line segments that are not used in this research.
Page 3, line(s) 87-91: Please explain why this study needs random re-sampling.
Page 3, line(s) 95-96: Please explain why the two regions use different models (GAMs and GLMs).
Page 4, line(s) 97-101: What kind of software and packages are these two models used?
Page 4, line(s) 97-101: What parameters are put into the two models?
Page 4, line(s) 97-101: How to convert Total catch to biomass needs to be clarified.
Page 4, line(s) 102-111: In this chapter, emphasis should be placed on completing the materials and methods of this study, and the descriptions that are unrelated to the materials and methods should be moved to the appropriate chapters.
Page 4, Table 2: The south study area only has two years of data (2013 and 2021). Such data is not suitable for trend analysis, only for differential analyses.
Page 5, Figure 2: The label of the y-axis needs to be clarified.
Page 5, Figure 2: The 95% CI of RAL is different from common sense. Please explain whether the model of RDA is valid.
Page 5, line(s) 131-132: It is recommended that the authors explain in detail what the Species codes represent.
Page 7, Figure 4: The south study area only has two years of data (2013 and 2021), which should not give readers meaningless information through invalid trend analysis.
Page 9, line(s) 184: Please confirm the correctness of the information on the line.
Page 9, line(s) 187-194: The 8-10 years required for those species to mature should be presented in the introduction.
Page 9, line(s) 195-202: The authors raised a critical question regarding sampling methods. At the end of this paragraph, it should state whether differences in sampling methods can be resolved by statistical analysis of the study.
Page 9, line(s) 217: Please synthesize the complete results in the final paragraph and explain the conclusions supported by the results.
Round 2
Reviewer 3 Report
1. I found that the authors had thought through and responded to every question raised by the reviewers.
2. Page 3, Figure 1: The information in the picture is relatively straightforward after the authors' revisions. However, I still need help understanding the significance of the 15 lines running from the upper left corner to the coastline on this Figure. Then, Falkland Islands Conservation Zones recommends that authors add to the image on the right, as the image on the left is incomplete. By the way, the authors should reconfirm the units of the label of the X-axis and Y-axis (currently, only the orientation (S, N) is seen).
3. Through the authors' explanations, I have understood the author's general approach and analysis purpose of using GAMs. However, GAMs are primarily used in multivariate analysis. But, the authors only have one variate (year). In addition, the missing value interval from 2013 to 2018 is too long. The data in Figure 2 shows that many species have a significant variation in the same year. The variation between years is likely to be insignificant compared to the former. This phenomenon dramatically challenges the explanatory power of the model. The authors should step up their discussion of the necessity and uncertainties of these models.
